# From MLP to NeoMLP:
# Leveraging Self-Attention for Neural Fields

## Abstract

Neural fields (NeFs) have recently emerged as a state-of-the-art method for encoding spatio-temporal signals of various modalities. Despite the success of NeFs in reconstructing individual signals, their use as representations in downstream tasks, such as classification or segmentation, is hindered by the complexity of the parameter space and its underlying symmetries, in addition to the lack of powerful and scalable conditioning mechanisms. In this work, we draw inspiration from the principles of connectionism to design a new architecture based on MLPs, which we term *Neo*MLP. We start from an MLP, viewed as a graph, and transform it from a multi-partite graph to a *complete graph* of input, hidden, and output nodes, equipped with *high-dimensional features*. We perform message passing on this graph and employ weight-sharing via *self-attention* among all the nodes. *Neo*MLP has a built-in mechanism for conditioning through the hidden and output nodes, which function as a set of latent codes, and as such, *Neo*MLP can be used straightforwardly as a conditional neural field. We demonstrate the effectiveness of our method by fitting high-resolution signals, including multi-modal audio-visual data. Furthermore, we fit datasets of neural representations, by learning instance-specific sets of latent codes using a single backbone architecture, and then use them for downstream tasks, outperforming recent state-of-the-art methods.

## 1 Introduction

The omnipresence of neural networks in the last decade has recently given rise to neural fields (NeFs) (*cf.* Xie et al. (2022)) as a powerful and scalable method to encode continuous signals of various modalities. These range from shapes (Park et al., 2019), scenes (Mildenhall et al., 2020), and images, (Sitzmann et al., 2020), to physical fields (Kofinas et al., 2023), CT scans (Papa et al., 2023; de Vries et al., 2024), and partial differential equations (Yin et al., 2022; Knigge et al., 2024). Consequently, the popularity of NeFs has spurred interest in *neural representations*, *i.e.* using NeFs as representations for downstream tasks.

Existing neural representations, however, suffer from notable drawbacks. Representations based on unconditional neural fields, *i.e.* independent multi-layer perceptrons (MLPs) fitted on each signal, are subject to parameter symmetries (Hecht-Nielsen, 1990), which lead to extremely poor performance in downstream tasks if left unattended (Navon et al., 2023). Many recent works (Navon et al., 2023; Zhou et al., 2023; Kofinas et al., 2024; Lim et al., 2024a; Papa et al., 2024) have proposed architectures that respect the underlying symmetries; the performance, however, leaves much to be desired. Another line of works (Park et al., 2019; Dupont et al., 2022) has proposed conditional neural fields with a single latent code per signal that modulates the activations of a shared MLP through concatenation, FiLM (Perez et al., 2018), or hypernetworks (Ha et al., 2016), while, recently, other works (Sajjadi et al., 2022; Wessels et al., 2024) have proposed set-latent conditional neural fields—conditional neural fields with a set of latent codes—that condition the signal through attention (Vaswani et al., 2017). Whilst the study of Rebain et al. (2022) showed that set-latent neural fields outperform single latent code methods as conditioning mechanisms, existing set-latent neural fields are based on cross-attention, which limits their scalability and expressivity: coordinates are only used as queries in attention, and cross-attention is limited to a single layer.

We argue that many of these drawbacks stem from the lack of a unified native architecture that integrates the necessary properties of neural representations and eliminates the shortcomings of

current approaches. To address these concerns, we draw inspiration from *connectionism* and the long history of MLPs to design a new architecture that functions as a standard machine learning model—akin to an MLP—as well as a conditional neural field. The paradigm of neural networks, from the early days of Perceptron (McCulloch & Pitts, 1943), to MLPs with hidden neurons trained with backpropagation (Rumelhart et al., 1986), to modern transformers (Vaswani et al., 2017), shares the connectionist principle: cognitive processes can be described by interconnected networks of simple and often uniform units.

This principle is lacking from current conditional neural field architectures, since conditioning is added to the network as an ad-hoc mechanism. In contrast, motivated by this principle, we take a closer look at MLPs; more specifically, we look at MLPs as a graph— similar to a few recent works (Kofinas et al., 2024; Lim et al., 2024a; Nikolentzos et al., 2024)— and design a novel architecture that operates on this graph using message passing. First, we convert the graph from a multi-partite graph to a fully-connected graph with self-edges. Instead of using edge-specific weights, we employ weight-sharing via self-attention among all the nodes. We initialize the hidden and output nodes with noise and optimize their values with backpropagation. Finally, we use high-dimensional features for all nodes to make self-attention and the network as a whole more scalable.

We make the following contributions. First, we propose a new architecture, which we term *Neo*MLP, by viewing MLPs as a graph, and convert this graph to a *complete graph* of input, hidden, and output nodes with *high-dimensional features*. We employ message passing on that graph through self-attention among the input, hidden, and output nodes. The hidden and output nodes can be used as a learnable set of latent codes, and thus, our method can function as a conditional neural field. We introduce new neural representations that use sets of latent codes for each signal, which we term $\nu$-reps, as well as datasets of neural representations, which we term $\nu$-sets. We fit datasets of signals using a single backbone architecture, and then use the latent codes for downstream tasks, outperforming recent state-of-the-art methods. We also demonstrate the effectiveness of our method by fitting high-resolution audio and video signals, as well as multi-modal audio-visual data.

## 1.1 BACKGROUND ON NEURAL FIELDS

Neural fields (NeFs), often referred to as Implicit Neural Representations (INRs), are a class of neural networks that parameterize fields using neural networks (*cf.* Xie et al. (2022)). In their simplest form, they are MLPs that take as input a single coordinate (*e.g.* an $x - y$ coordinate) and output the field value for that coordinate (*e.g.* an RGB value). By feeding batches of coordinates to the network, and training to reconstruct the target values with backpropagation, the neural field learns to encode the target signal, without being bound to a specific resolution.

Conditional neural fields introduce a conditioning mechanism to neural fields through latent variables, often referred to as *latent codes*. This conditioning mechanism can be used to encode instance-specific information (*e.g.* encode a single image) and disentangle it from the backbone architecture, which now carries dataset-wide information.

## 2 NEOMLP

### 2.1 FROM MLP TO NEOMLP

We begin the exposition of our method with MLPs, since our architecture is influenced by MLPs and builds on them. Without loss of generality, a multi-layer perceptron takes as input a set of scalar variables $\{x_i\}_{i=1}^I, x_i \in \mathbb{R}$, coalesced into a single high-dimensional array $\mathbf{x} \in \mathbb{R}^I$. Through a series of non-linear transformations, the input array is progressively transformed into intermediate (hidden) representations, with the final transformation leading to the output array $\mathbf{y} \in \mathbb{R}^O$.

Akin to other recent works (Kofinas et al., 2024; Lim et al., 2024b; Nikolentzos et al., 2024), we look at an MLP as a graph; an MLP is an $L + 1$-partite graph, where $L$ is the number of layers. The nodes represent the input, hidden, and output neurons, and have scalar features that correspond to individual inputs, the hidden features at each layer, and the individual outputs, respectively. We perform message passing on that graph, after making it more amenable for learning. First, we convert the connectivity graph from an $L + 1$-partite graph to a fully-connected graph with self-edges. Since the forward pass now includes message passing from all nodes to all nodes at each step, we create learnable

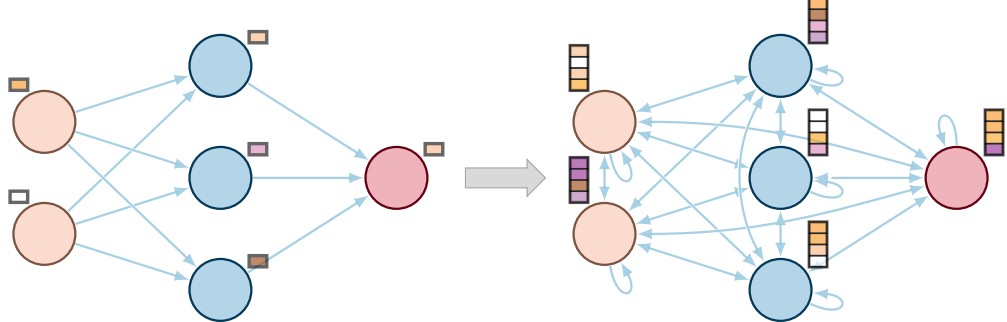

Figure 1: The connectivity graphs of MLP and *Neo*MLP. *Neo*MLP performs message passing on the MLP graph. Going from MLP to *Neo*MLP, we use a fully connected graph and high-dimensional node features. In *Neo*MLP, the traditional notion of layers of neurons, as well as the asynchronous layer-wise propagation, cease to exist. Instead, we use synchronous message passing with weight-sharing via self-attention among all the nodes. *Neo*MLP has three types of nodes: input, hidden, and output nodes. The input is fed to *Neo*MLP through the input nodes, while the output nodes capture the output of the network.

parameters for the initial values of the hidden and output node features. We initialize them with Gaussian noise, and optimize their values with backpropagation, simultaneously with the network parameters. Next, we observe that having dedicated edge-specific weights for all node pairs would result in an intractable spatial complexity. As such, in order to reduce the memory footprint, we follow the standard practice of graph neural networks and Transformers (Vaswani et al., 2017), and employ weight-sharing between the nodes, specifically via self-attention. In other words, the weights for each node pair are computed as a function of the incoming and outgoing node features, in conjunction with weights that are shared across nodes. As a by-product of the self-attention mechanism, which is permutation invariant, we use node-specific embeddings that allow us to differentiate between different nodes. Finally, instead of having scalar node features, we increase the dimensionality of node features, which makes self-attention more scalable and expressive.

We show the connectivity graph of *Neo*MLP and its conversion from a standard MLP in Figure 1. We also show the equations of the forward pass for a single layer of an MLP and a simplified version of *Neo*MLP (without softmax normalization, scaling, or multi-head attention) in Equation (1).

$$
\begin{aligned}
\text{MLP:} \quad & \mathrm{h}_i^{(l)} = \sum_j \overbrace{\mathrm{W}_{ij}^{(l)}}^{} \, \mathrm{h}_j^{(l-1)} \\
\textit{Neo}\text{MLP:} \quad & \mathbf{h}_i^{(l)} = \sum_j \overbrace{\left(\mathbf{W}_Q^{(l)} \mathbf{h}_i^{(l-1)}\right)^{\top} \mathbf{W}_K^{(l)} \mathbf{h}_j^{(l-1)} \mathbf{W}_V^{(l)}}^{} \, \mathbf{h}_j^{(l-1)}
\end{aligned}
\tag{1}
$$

We note that throughout this work, we retain the nomenclature of input, hidden, and output nodes, but repurpose them for *Neo*MLP. More specifically, these nodes refer to the connectivity graph of *Neo*MLP, *i.e.* the graph on which we perform message passing, shown in Figure 1, and not its computational graph, which would include layers of all the nodes. The input is fed to *Neo*MLP through the input nodes before any information propagation, while the output nodes are the ones that will capture the output of the network, after a number of message passing layers. Every other node that is not used for input or output is a hidden node. The number of hidden nodes in *Neo*MLP does not need to correspond one-to-one to the MLP hidden nodes.

## 2.2 NEOMLP ARCHITECTURE

After establishing the connection with MLPs, we now discuss the architecture of our method in detail. The inputs comprise a set of scalar variables $\{x_i\}_{i=1}^I, x_i \in \mathbb{R}$. We employ random Fourier features (Tancik et al., 2020) as a non-learnable method to project each scalar input (each dimension separately) to a high-dimensional space $\mathbb{R}^{D_{\text{RFF}}}$. This is followed by a linear layer that projects it to $\mathbb{R}^D$. We then add learnable positional embeddings to the inputs. These embeddings are required for the model to differentiate between input variables, since self-attention is a permutation invariant operation. We use similar learnable embeddings for each scalar output dimension (referred to as

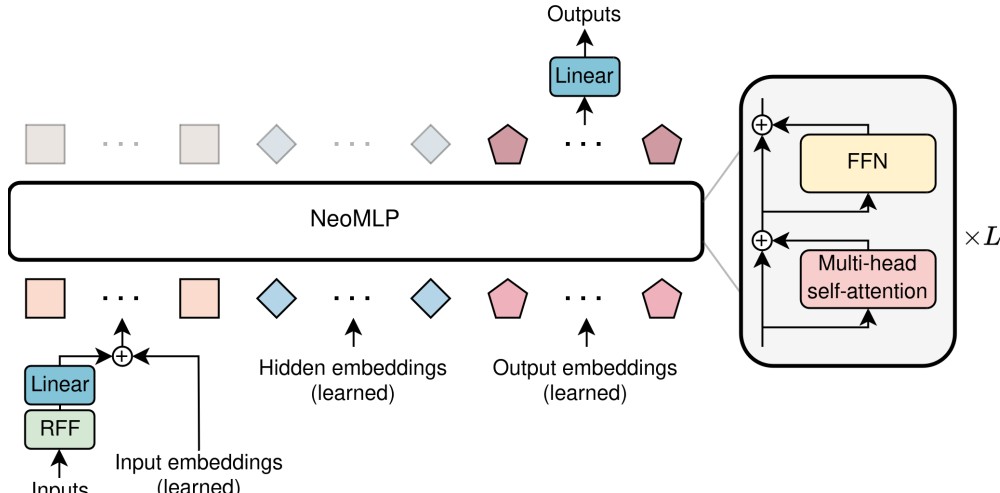

Figure 2: The architecture of *Neo*MLP. We pass each input dimension through an RFF layer followed by a linear layer, and then add individual input embeddings to each input. The transformed inputs, alongside the embeddings for the hidden and output nodes, comprise the inputs to *Neo*MLP. *Neo*MLP has $L$ layers of residual self-attention and non-linear transformations. We capture the output that corresponds to the output nodes and pass it through a linear layer to get the final output of the network.

output embeddings), as well as $H$ learnable embeddings for each hidden node (referred to as hidden embeddings), where $H$ is chosen as a hyperparameter. We concatenate the transformed inputs with the hidden and output embeddings along the node (token) dimension, before feeding them to *Neo*MLP. We denote the concatenated tokens as $\mathbf{T}^{(0)} \in \mathbb{R}^{(I+H+O)\times D}$, where $O$ is the number of output dimensions. The input, hidden, and output embeddings are initialized with Gaussian noise. We use a variance $\sigma_i^2$ for the input embeddings and $\sigma_o^2$ for the hidden and output embeddings; both are chosen as hyperparameters.

Each *Neo*MLP layer comprises a multi-head self-attention layer among the tokens, and a feed-forward network that non-linearly transforms each token independently. The output of each layer consists of the transformed tokens $\mathbf{T}^{(l)} \in \mathbb{R}^{(I+H+O)\times D}$. We use pre-LN transformer blocks (Xiong et al., 2020), but omit LayerNorm (Ba et al., 2016), since we observed it does not lead to better performance or faster convergence. This also makes our method conceptually simpler. Thus, a *Neo*MLP layer is defined as follows:

$$\widetilde{\mathbf{T}}^{(l)} = \mathbf{T}^{(l-1)} + \text{SelfAttention}\left(\mathbf{T}^{(l-1)}\right) \tag{2}$$

$$\mathbf{T}^{(l)} = \widetilde{\mathbf{T}}^{(l)} + \text{FeedForwardNetwork}\left(\widetilde{\mathbf{T}}^{(l)}\right) \tag{3}$$

We explore different variants of self-attention and find that linear attention (Katharopoulos et al., 2020; Shen et al., 2021) performs slightly better and results in a faster model, while simultaneously requiring fewer parameters. Specifically, we use the version of Shen et al. (2021) from a publicly available implementation of linear attention[1].

After $L$ *Neo*MLP layers, we only keep the final tokens that correspond to the output embeddings, and pass them through a linear layer that projects them back to scalars. We then concatenate all outputs together, which gives us the final output array $\mathbf{y} \in \mathbb{R}^O$. The full pipeline of our method is shown in

---

[1]https://github.com/lucidrains/linear-attention-transformer

Figure 2, while the forward pass is mathematically described as follows:

$$\mathbf{i}_i = \text{Linear}(\text{RFF}(x_i)) + \text{InputEmbedding}(i), \qquad i \in \{1, \ldots, I\}, \mathbf{i}_i \in \mathbb{R}^D \tag{4}$$

$$\mathbf{h}_j = \text{HiddenEmbedding}(j), \qquad j \in \{1, \ldots, H\}, \mathbf{h}_j \in \mathbb{R}^D \tag{5}$$

$$\mathbf{o}_k = \text{OutputEmbedding}(k), \qquad k \in \{1, \ldots, O\}, \mathbf{o}_k \in \mathbb{R}^{O \times D} \tag{6}$$

$$\mathbf{T}^{(0)} = \left[ \{\mathbf{i}_i\}_{i=1}^I, \{\mathbf{h}_j\}_{j=1}^H, \{\mathbf{o}_k\}_{k=1}^O \right], \qquad \mathbf{T}^{(0)} \in \mathbb{R}^{(I+H+O) \times D} \tag{7}$$

$$\mathbf{T}^{(l)} = \text{NeoMLPLayer}\left( \mathbf{T}^{(l-1)} \right), \qquad l \in \{1, \ldots, L\}, \mathbf{T}^{(l)} \in \mathbb{R}^{(I+H+O) \times D} \tag{8}$$

$$\mathbf{y} = \text{Linear}\left( \mathbf{T}^{(L)}_{I+H:I+H+O} \right), \qquad \mathbf{y} \in \mathbb{R}^{O \times 1} \tag{9}$$

## 2.3 NeoMLP as an auto-decoding conditional neural field

One of the advantages of our method is its adaptability, since it has a built-in mechanism for conditioning, through the hidden and output embeddings. In the context of neural fields, this mechanism enables our method to function as an auto-decoding conditional neural field (Park et al., 2019), while the embeddings can be used as neural representations for downstream tasks, shown schematically in Figure 3. We refer to these representations as $\nu$-reps (nu-reps), and similarly, we refer to the datasets of neural representations obtained with our method as $\nu$-sets (nu-sets).

As a conditional neural field, the *Neo*MLP backbone encodes the neural field parameters, while the latent variables, *i.e.* the hidden and output embeddings, encode instance-specific information. Each instance (*e.g.* each image in an image dataset) is represented with its own set of latent codes $\mathbf{Z}_n = \left[ \{\mathbf{h}_j^n\}_{j=1}^H, \{\mathbf{o}_k^n\}_{k=1}^O \right]$. We optimize the latent codes for a particular signal by feeding them to the network as inputs alongside a coordinate $\mathbf{x}_p^{(n)}$, compute the field value $\hat{\mathbf{y}}_p^{(n)}$ and the reconstruction loss, and backpropagate the loss to $\mathbf{Z}_n$ to take one optimization step.

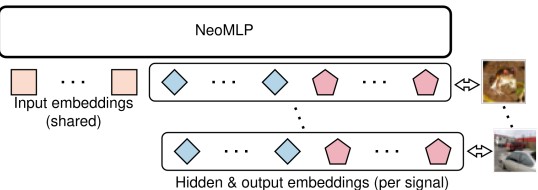

Figure 3: The hidden and output embeddings constitute a set of latent codes for each signal, and can be used as neural representations for downstream tasks. We term these neural representations as $\nu$-reps, and the datasets of neural representations as $\nu$-sets.

Our method operates in two distinct stages: fitting and finetuning. During fitting, our goal is to optimize the backbone architecture, *i.e.* the parameters of the model. We sample latent codes for all the signals of a fitting dataset and optimize them simultaneously with the backbone architecture. When the fitting stage is complete, after a predetermined set of epochs, we freeze the parameters of the backbone architecture and discard the latent codes. Then, during finetuning, given a new signal, we sample new latent codes for it and optimize them to minimize the reconstruction error for a number of epochs. We finetune the training, validation, and test sets of the downstream task from scratch, even if we used the training set to fit the model, in order to make the distance of representations between splits as small as possible.

In both the fitting and the finetuning stage, we sample completely random points from random signals. This ensures $i.i.d.$ samples, and speeds up the training of our method. During the fitting stage, we also sample points *with replacement*, as we observed a spiky behaviour in the training loss otherwise. We provide the detailed algorithms of the fitting and the finetuning stage in Algorithms 1 and 2 in Appendix A, respectively. We provide further implementation details in Appendix D.

## 2.4 Using $\nu$-reps for downstream tasks

After finetuning neural representations, our goal is to use them in downstream tasks, *e.g.* to train a downstream model for classification or segmentation. Our $\nu$-reps comprise a set of latent codes for each signal, corresponding to the finetuned hidden and output embeddings. While the space of $\nu$-reps is subject to permutation symmetries, which we discuss in Appendix B, we use a simple downstream model that first concatenates and flattens the hidden and output embeddings in a single vector, and

then process it with an MLP. We leave more elaborate methods that exploit the inductive biases present in $\nu$-reps for future work.

## 3 EXPERIMENTS

We gauge the effectiveness of our approach by fitting individual high-resolution signals, as well as datasets of signals. We also evaluate our method on downstream tasks on the fitted datasets. We refer to the appendix for more details. The code is included in the supplementary material and will be open-sourced to facilitate reproduction of the results.

### 3.1 FITTING HIGH-RESOLUTION SIGNALS

First, we evaluate our method at fitting high-resolution signals. We compare our method against Siren (Sitzmann et al., 2020), an MLP with sinusoidal activations, RFFNet (Tancik et al., 2020), an MLP with random Fourier features and ReLU activations, and SPDER (Shah & Sitawarin, 2024), an MLP with sublinear damping activations combined with sinusoids. Our goal is to assess the effectiveness of our method in signals of various modalities, and especially in multimodal signals, which have been underexplored in the context of neural fields. Hence, we choose signals that belong to two different modalities, namely an audio clip and a video clip, as well as a multi-modal signal, namely video with audio.

For audio, we follow Siren (Sitzmann et al., 2020) and use the first 7 seconds from Bach's cello suite No. 1 in G Major: Prelude. The audio clip is sampled at $44.1$ kHz, resulting in 308,700 points. For video, we use the "bikes" video from the `scikit-video` Python library, available online[2]. This video clip lasts for 10 seconds and is sampled at 25 fps, with a spatial resolution of $272 \times 640$, resulting in 43,520,000 points. Finally, we explore multimodality using the "Big Buck Bunny" video from `scikit-video`. This clip lasts for 5.3 seconds. The audio is sampled at $48$ kHz and has 6 channels. The original spatial resolution is $1280 \times 720$ at 25 fps. We subsample the spatial resolution by 2, which results in a resolution of $640 \times 360$. Overall, this results in 30,667,776 points (254,976 from audio and 30,412,800 from video).

**Training details**  For audio, we follow Siren (Sitzmann et al., 2020) and scale the time domain to $t \in [-100, 100]$ instead of $[-1, 1]$, to account for the high sampling rate of the signal. For the audio-visual data, we model the signal as $f : \mathbb{R}^3 \to \mathbb{R}^9$, *i.e.* we have 3 input dimensions $(x, y, t)$, and 9 output dimensions: 3 from video (RGB) and 6 from audio (6 audio channels). Similar to the audio clip, we also scale the time domain, which is now used as the time coordinate for both the audio and the video points. For the points corresponding to audio, we fill their $xy$ coordinates with zeros. Furthermore, since all points come from either the video or the audio modality, we fill the output dimensions that correspond to the other modality with zeros. Finally, during training, we mask these placeholder output dimensions, *i.e.* we compute the loss for the video coordinates using only the RGB outputs, and the loss for the audio coordinates using only the 6-channel audio outputs.

To ensure fairness, for every signal, *Neo*MLP has approximately the same number of parameters as the baselines. We describe the architecture details for each experiment in Appendix E. We show the results in Table 1, measuring the reconstruction PSNR. We observe that *Neo*MLP comfortably outperforms the baselines in all three signals. Interestingly, the performance gap is increased in the more difficult setup of multimodal data, which suggests the suitability of our method for multimodal signals. We hypothesize that this can be attributed to our method's ability to learn faster from minibatches with $i.i.d.$ elements, which is something we observed empirically during training and hyperparameter tuning. We visualize example frames for the video clips in Figure 4, and in Figure 6 in Appendix G. We provide further qualitative results in Appendix G and include reconstructions of all signals in the supplementary material.

### 3.2 FITTING $\nu$-SETS & DOWNSTREAM TASKS ON $\nu$-SETS

Next, we evaluate our method on fitting $\nu$-sets, *i.e.* fitting datasets of neural representations of signals with *Neo*MLP, as well as performing downstream tasks on $\nu$-sets. We compare our method against

---

[2] https://www.scikit-video.org/stable/datasets.html

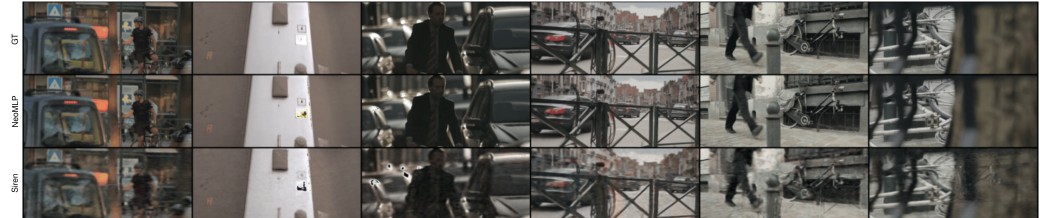

Figure 4: Examples frames from fitting the "bikes" video clip. The first row shows the groundtruth, while the second and the third row show the reconstructions obtained using *Neo*MLP and Siren, respectively. We observe that *Neo*MLP learns to reconstruct the video with much greater fidelity.

Table 1: Performance on fitting high resolution signals. We report the PSNR (higher is better).

| Method | Dataset | | | |
| --- | --- | --- | --- | --- |
| | Bach | Bikes | Big Buck Bunny | |
| | | | Audio | Video |
| RFFNet (Tancik et al., 2020) | 54.62 | 27.00 | 32.88 | 24.59 |
| Siren (Sitzmann et al., 2020) | 51.65 | 37.02 | 31.55 | 24.82 |
| SPDER (Shah & Sitawarin, 2024) | 48.06 | 33.82 | 28.45 | 20.90 |
| *Neo*MLP (ours) | **54.71** | **39.06** | **39.00** | **34.17** |

Functa (Dupont et al., 2022), DWSNet (Navon et al., 2023), Neural Graphs (Kofinas et al., 2024), and Fit-a-NeF (Papa et al., 2024). Functa is a conditional neural field that uses an MLP backbone and conditioning by bias modulation. DWSNet, Neural Graphs, and Fit-a-NeF, on the other hand, are equivariant downstream models for processing datasets of unconditional neural fields. For these three methods, the process of creating datasets of neural representations corresponds to fitting separate MLPs for each signal in a dataset, a process that is independent of the downstream models themselves. Since these methods have the step of generating the neural datasets in common, we use shared datasets for these methods, provided by Fit-a-NeF.

We consider three datasets, namely MNIST (LeCun et al., 1998), CIFAR10 (Krizhevsky et al., 2009), and ShapeNet10 (Chang et al., 2015). We evaluate reconstruction quality for MNIST and CIFAR10 with PSNR, and for ShapeNet with IoU. For CIFAR10, we follow the setup of Functa (Dupont et al., 2022), and use 50 augmentations for all training and validation images during finetuning. For all datasets, we only use the training set as a fitting set, since this closely mimics the real-world conditions for auto-decoding neural fields, namely that test set data can appear after the backbone is frozen, and should be finetuned without changing the backbone.

After fitting the neural datasets, we optimize the downstream model for the downstream tasks, which corresponds to classification for MNIST, CIFAR10, and ShapeNet10. We perform a hyperparameter search for *Neo*MLP to find the best downstream model. Specifically, we use Bayesian hyperparameter search from Wandb (Biewald, 2020) to find the best performing hyperparameters for CIFAR10, and reuse these hyperparameters for all datasets.

While neural datasets can easily reach excellent reconstruction quality, it is often at the expense of representation power. This was shown in the case of unconditional neural fields by Papa et al. (2024), where optimal downstream performance was often achieved with medium quality reconstructions. Since our goal in this experiment is to optimize the performance of neural representations in downstream tasks, we report the reconstruction quality of the models that achieved the best downstream performance.

We report the results in Table 2. We observe that *Neo*MLP comfortably outperforms DWSNet (Navon et al., 2023), Neural Graphs (Kofinas et al., 2024) and Fit-a-NeF (Papa et al., 2024), *i.e.* all methods that process unconditional neural fields, both in terms of representation quality and downstream performance. Further, these two quantities seem to be positively correlated for *Neo*MLP, in contrast to

Table 2: Performance on fitting neural datasets and downstream classification for neural datasets. Experiments on MNIST, CIFAR10, and ShapeNet10. Results from methods marked with † were taken from Fit-a-NeF (Papa et al., 2024). The | symbols that appear above and below a number denote that this number is shared for these three methods. For classification, we run the experiments for 3 random seeds and report the mean and standard deviation.

| Method | MNIST | | CIFAR10 | | ShapeNet | |
|---|---|---|---|---|---|---|
| | PSNR ($\uparrow$) | Accuracy (%) | PSNR ($\uparrow$) | Accuracy (%) | IoU ($\uparrow$) | Accuracy (%) |
| Functa (Dupont et al., 2022) | 33.07 | $98.73_{\pm 0.05}$ | 31.90 | $68.30_{\pm 0.00}$ | 0.434 | $95.23_{\pm 0.13}$ |
| DWSNet (Navon et al., 2023) † | | $85.70_{\pm 0.60}$ | | $44.01_{\pm 0.48}$ | | $91.06_{\pm 0.25}$ |
| Neural Graphs (Kofinas et al., 2024) † | 14.66 | $92.40_{\pm 0.30}$ | 20.45 | $44.11_{\pm 0.20}$ | 0.559 | $90.31_{\pm 0.15}$ |
| Fit-a-NeF (Papa et al., 2024) † | | $96.40_{\pm 0.11}$ | | $39.83_{\pm 1.70}$ | | $82.96_{\pm 0.02}$ |
| *Neo*MLP (ours) | **33.98** | $\mathbf{98.78_{\pm 0.04}}$ | **33.16** | $\mathbf{73.40_{\pm 0.12}}$ | **0.934** | $\mathbf{95.30_{\pm 0.08}}$ |

the findings of Papa et al. (2024) for unconditional neural fields. Our method also outperforms Functa (Dupont et al., 2022) on all three datasets regarding the classification accuracy, while maintaining an excellent reconstruction quality.

## 3.3 ABLATION STUDIES

**Importance of hyperparameters** We perform a large ablation study to assess the importance of the latent codes, and the impact of the duration of fitting and finetuning to the quality of reconstruction and representation power. Specifically, we run two studies on CIFAR10; the first study monitors the number and the dimensionality of the latent codes, as well as the number of finetuning epochs. The second study monitors the number and the dimensionality of the latent codes, as well as the number of fitting epochs. In both studies, all other hyperparameters are fixed. We report the fitting PSNR, the test PSNR and the downstream accuracy. We summarize our findings in Tables 3 and 4.

In both studies, we observe that increasing the number of latents and their dimensionality also increases the reconstruction quality. However, the higher number of latents seems to lead to decreased downstream performance. Furthermore, we notice that increasing the number of finetuning epochs also increases the test PSNR and accuracy. Finally, somewhat surprisingly, while fitting for more epochs leads to noticeably better fitting PSNR, this translates to negligible gain in the test PSNR and accuracy, and even degrades performance in some cases.

Table 3: Ablation study on the importance of the number of latents, the dimensionality of the latents, and the number of finetuning epochs. The backbone is fitted for 50 epochs. Experiment on CIFAR10; no augmentations are used in this study.

| Num. latents | Latent dim. | Fit PSNR ($\uparrow$) | Finetune for 5 epochs | | Finetune for 10 epochs | |
|---|---|---|---|---|---|---|
| | | | Test PSNR ($\uparrow$) | Accuracy (%) | Test PSNR ($\uparrow$) | Accuracy (%) |
| 6 | 64 | 27.04 | 24.67 | 51.23 | 26.00 | 50.86 |
| | 128 | 30.01 | 26.46 | 53.30 | 28.41 | 53.25 |
| | 256 | 33.10 | 28.17 | 53.76 | 30.82 | 54.52 |
| | 512 | 37.49 | **30.89** | **54.66** | **34.98** | **56.23** |
| 14 | 64 | 30.58 | 26.28 | 49.36 | 28.58 | 49.69 |
| | 128 | 34.59 | 28.34 | 50.74 | 31.52 | 51.28 |
| | 256 | 37.65 | 29.63 | 53.35 | 33.70 | 54.06 |
| | 512 | **39.30** | 30.77 | 53.26 | 33.99 | 53.65 |

**Further ablations** We perform more ablation experiments regarding the number of layers and hidden latents, and the importance of RFF and layer normalization. We report the results in Appendix J.

## 4 RELATED WORK

**Neural representations** An increasingly large body of works (Navon et al., 2023; Zhou et al., 2023; Kofinas et al., 2024; Lim et al., 2024a; Papa et al., 2024; Tran et al., 2024; Kalogeropoulos et al., 2024) has proposed downstream methods that process datasets of unconditional neural fields, *i.e.* the parameters and the architectures of MLPs. They are all addressing the parameter symmetries present in MLPs, and while the performance of such methods is constantly increasing, it still leaves much

Table 4: Ablation study on the importance of the number of latents, the dimensionality of the latents, and the number of fitting epochs. The latents are finetuned for 5 epochs. Experiment on CIFAR10; no augmentations are used in this study.

| Num. latents | Latent dim. | Fit 20 epochs | | | Fit 50 epochs | | |
|---|---|---|---|---|---|---|---|
| | | Fit PSNR ($\uparrow$) | Test PSNR ($\uparrow$) | Accuracy (%) | Fit PSNR ($\uparrow$) | Test PSNR ($\uparrow$) | Accuracy (%) |
| 6 | 64 | 25.68 | 24.68 | 51.03 | 27.04 | 24.67 | 51.23 |
| | 128 | 28.05 | 26.40 | 52.67 | 30.01 | 26.46 | 53.30 |
| | 256 | 30.04 | 28.17 | 54.56 | 33.10 | 28.17 | 53.76 |
| | 512 | 33.91 | 30.84 | **55.14** | 37.49 | **30.89** | 54.66 |
| 14 | 64 | 28.34 | 26.18 | 49.67 | 30.58 | 26.28 | 49.36 |
| | 128 | 31.63 | 28.03 | 52.12 | 34.59 | 28.34 | 50.74 |
| | 256 | 33.02 | 29.24 | 53.52 | 37.65 | 29.63 | 53.35 |
| | 512 | 31.94 | 30.54 | 54.42 | **39.30** | 30.77 | 53.26 |

to be desired. Closer to our work is another body of works (Park et al., 2019; Dupont et al., 2022; Sajjadi et al., 2022; Chen & Wang, 2022; Zhang et al., 2023; 2024; Wessels et al., 2024) that proposes neural representations through conditional neural fields. Of those, Sajjadi et al. (2022); Zhang et al. (2023); Wessels et al. (2024) have proposed set-latent conditional neural fields that condition the signal through attention (Vaswani et al., 2017). Zhang et al. (2023) proposed 3DShape2VecSet, an architecture that employs cross-attention and self-attention to encode shapes into sets of latent vectors and decode them. Our method differs from this method, since it does not rely on cross-attention to fully encode a coordinate in a set of latents. Instead, it employs self-attention, which allows for better information propagation and enables the model to scale to multiple layers.

**MLPs as graphs** A few recent works (Kofinas et al., 2024; Lim et al., 2024a;b; Nikolentzos et al., 2024; Kalogeropoulos et al., 2024) have viewed neural networks as graphs and proposed methods that leverage the graph structure. Kofinas et al. (2024) focus on the task of processing the parameters of neural networks and represent neural networks as computational graphs of parameters. Their method includes applications to downstream tasks on neural fields. Lim et al. (2024b) investigate the impact of parameter symmetries, and introduce new neural network architectures that have reduced parameter space symmetries. Nikolentzos et al. (2024) show that MLPs can be formalized as GNNs with asynchronous message passing, and propose a model that employs synchronous message passing on a nearly complete graph. Similar to this work, we use a complete graph and employ a synchronous message passing scheme. In contrast to this work, we employ weight-sharing via self-attention and high-dimensional node features. Further, we focus on NeF applications instead of tabular data, and explore conditioning via the hidden and output embeddings.

## 5 CONCLUSION

In this work, we presented *Neo*MLP, a novel architecture inspired by the principles of connectionism and the graph perspective of MLPs. We perform message passing on the graph of MLPs, after transforming it to a complete graph of input, hidden, and output nodes equipped with high-dimensional features. We also employ weight-sharing through self-attention among all the nodes. *Neo*MLP is a transformer architecture that uses individual input and output dimensions as tokens, along with a number of hidden tokens. We also introduced new neural representations based on the hidden and output embeddings, as well as datasets of neural representations. Our method achieves state-of-the-art performance in fitting high-resolution signals, including multimodal audio-visual data, and outperforms state-of-the-art methods in downstream tasks on neural representations.

**Limitations** Our $\nu$-reps are subject to permutation symmetries, indicating that inductive biases can be leveraged to increase downstream performance. Namely, while the output embeddings are already ordered, as they correspond to individual outputs, the hidden embeddings are subject to permutation symmetries. Future work can explore more elaborate methods based on set neural networks, such as Deep Sets (Zaheer et al., 2017), that exploit the inductive biases present in $\nu$-reps. Further, the latent codes used in $\nu$-reps, namely the hidden and output embeddings, carry global information. Instilling locality in latent codes can be useful for fine-grained downstream tasks, such as segmentation. Future work can explore equivariant neural fields (Wessels et al., 2024), which would localize the latent codes by augmenting them with positions or orientations.

## REPRODUCIBILITY STATEMENT

We use publicly available data and datasets, which are described in Section 3. The code is included in the supplementary material. Equations (2) and (4) mathematically describe our method. Further, we describe the algorithms for fitting and finetuning *Neo*MLP in Algorithms 1 and 2, respectively. We report details regarding the implementation in Appendix D, dataset details in Appendix F, and details about the hyperparameters used in each experiment in Appendix E.

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

## A   FITTING AND FINETUNING $\nu$-SETS

---

**Algorithm 1** Fit *Neo*MLP as a conditional neural field

---

**Require:** Randomly initialized backbone network $\mathbf{f}_\Theta$

**Require:** Fitting dataset: $\mathcal{D}_{\text{fit}} = \left\{ \left\{ \mathbf{x}_p^{(n)}, \mathbf{y}_p^{(n)} \right\}_{p=1}^{P_n} \right\}_{n=1}^{N_{\text{fit}}}$   $\triangleright$ $N_{\text{fit}}$ signals, Coordinate $\mathbf{x}_p^{(n)} \in \mathbb{R}^I$

$\triangleright$ Field value $\mathbf{y}_p^{(n)} \in \mathbb{R}^O$

**Require:** Randomly initialized latents: $\mathcal{Z}_{\text{fit}} = \{\mathbf{Z}_n\}_{n=1}^{N_{\text{fit}}}$

**Require:** Initialized optimizer: $O_{\text{fit}}$   $\triangleright$ Adam (Kingma & Ba, 2015)

**Require:** Number of fitting epochs $E$

**Require:** Fitting minibatch size $B$   $\triangleright$ Number of *points* per minibatch

$P \leftarrow \sum_{n=1}^{N_{\text{fit}}} P_n$   $\triangleright$ Total number of points in the dataset

$M \leftarrow \lfloor \frac{P}{B} \rfloor$   $\triangleright$ Number of iterations per epoch. We drop incomplete minibatches

**function** FITNEOMLP

    **for** epoch $\in \{1, \ldots, E\}$ **do**

        **for** iteration $\in \{1, \ldots, M\}$ **do**

            Sample point indices $\mathbb{P} = \{p_b\}_{b=1}^B$

            Sample signal indices $\mathbb{S} = \{n_b\}_{b=1}^B$   $\triangleright$ Sample $\mathbb{P}$ and $\mathbb{S}$ *with replacement*

            $\mathcal{B} \leftarrow \left\{ \mathbf{x}_{p_b}^{(n_b)}, \mathbf{y}_{p_b}^{(n_b)}, \mathbf{Z}_{n_b} \right\}_{b=1}^B$

            $\hat{\mathbf{y}}_{p_b}^{(n_b)} \leftarrow \mathbf{f}_\Theta \left( \mathbf{x}_{p_b}^{(n_b)}, \mathbf{Z}_{n_b} \right)$   $\triangleright$ In parallel $\forall\, b \in \{1, \ldots, B\}$

            $\mathcal{L} \leftarrow \frac{1}{B} \sum_{b=1}^B \left\| \mathbf{y}_{p_b}^{(n_b)} - \hat{\mathbf{y}}_{p_b}^{(n_b)} \right\|_2^2$

            $\Theta \leftarrow \Theta - O_{\text{fit}}(\nabla_\Theta \mathcal{L})$

            $\mathbf{Z}_{n_b} \leftarrow \mathbf{Z}_{n_b} - O_{\text{fit}}\left(\nabla_{\mathbf{Z}_{n_b}} \mathcal{L}\right)$   $\triangleright$ In parallel $\forall\, b \in \{1, \ldots, B\}$

        **end for**

    **end for**

    Freeze $\Theta$

    Discard $\mathcal{Z}_{\text{fit}}$

    **return** $\Theta$

**end function**

---

## B   NEOMLP SYMMETRIES

Our $\nu$-reps, and more specifically, the hidden embeddings, are subject to permutation symmetries. Intuitively, when we permute two hidden embeddings from a randomly initialized or a trained model, we expect the behaviour of the network to remain the same. In this section, we formalize the permutation symmetries present in our method. *Neo*MLP is a function $f : \mathbb{R}^{(I+H+O)\times D} \to \mathbb{R}^{(I+H+O)\times D}$ that comprises self-attention and feed-forward networks applied interchangeably for a number of layers, following Equations (2) and (3). As a transformer architecture, it is a permutation equivariant function. Thus, the following property holds: $f(\mathbf{PX}) = \mathbf{P}f(\mathbf{X})$, where $\mathbf{P}$ is a permutation matrix, and $\mathbf{X}$ is a set of tokens fed as input to the transformer.

Now consider the input to *Neo*MLP: $\mathbf{T}^{(0)} = \left[ \{\mathbf{i}_i\}_{i=1}^I, \{\mathbf{h}_j\}_{j=1}^H, \{\mathbf{o}_k\}_{k=1}^O \right], \mathbf{T}^{(0)} \in \mathbb{R}^{(I+H+O)\times D}$. We look at two cases of permutations, namely permuting only the hidden neurons, and permuting only the output neurons. The permutation matrix for the first case, *i.e.* permuting only the hidden neurons, is $\mathbf{P}_1 = \mathbf{I}_{I\times I} \oplus \mathbf{P}_{H\times H} \oplus \mathbf{I}_{O\times O}$, where $\mathbf{I}$ is the identity matrix, $\mathbf{P}_{H\times H}$ is a permutation matrix, and $\oplus$ denotes the direct sum operator, *i.e.* stacking matrix blocks diagonally, with zero matrices in the off-diagonal blocks. Each $\mathbf{P}_1$ corresponds to a permutation $\pi_1 \in S_H$.

Applying this permutation to $\mathbf{T}^{(0)}$ permutes only the hidden neurons:

$$\mathbf{P}_1 \mathbf{T}^{(0)} = \left[ \{\mathbf{i}_i\}_{i=1}^I, \left\{ \mathbf{h}_{\pi_1^{-1}(j)} \right\}_{j=1}^H, \{\mathbf{o}_k\}_{k=1}^O \right] \tag{10}$$

**Algorithm 2** Finetune *Neo*MLP as a conditional neural field

**Require:** Frozen backbone network $\mathbf{f}_\Theta$
**Require:** Train, validation, test datasets: $\mathcal{D}_{\text{train}}, \mathcal{D}_{\text{validation}}, \mathcal{D}_{\text{test}}$
**Require:** Randomly initialized latents: $\mathcal{Z}_{\text{train}}, \mathcal{Z}_{\text{validation}}, \mathcal{Z}_{\text{test}}$
**Require:** Initialized optimizers: $O_{\text{train}}, O_{\text{validation}}, O_{\text{test}}$      ▷ Adam (Kingma & Ba, 2015)
**Require:** Number of finetuning epochs $E'$
**Require:** Finetuning minibatch size $B'$
  **function** FINETUNENEOMLP
    **for** split $\in \{$train, validation, test$\}$ **do**
      $M_{\text{split}} \leftarrow \lceil \frac{\sum_{n=1}^{N_{\text{split}}} P_n}{B'} \rceil$
      **for** epoch $\in \{1, \ldots, E'\}$ **do**
        **for** iteration $\in \{1, \ldots, M_{\text{split}}\}$ **do**
          Sample point indices $\mathbb{P} = \{p_b\}_{b=1}^{B'}$
          Sample signal indices $\mathbb{S} = \{n_b\}_{b=1}^{B'}$     ▷ Sample $\mathbb{P}$ and $\mathbb{S}$ without replacement
          $\mathcal{B} \leftarrow \left\{ \mathbf{x}_{p_b}^{(n_b)}, \mathbf{y}_{p_b}^{(n_b)}, \mathbf{Z}_{n_b} \right\}_{b=1}^{B'}$
          $\hat{\mathbf{y}}_{p_b}^{(n_b)} \leftarrow \mathbf{f}_\Theta \left( \mathbf{x}_{p_b}^{(n_b)}, \mathbf{Z}_{n_b} \right)$     ▷ In parallel $\forall\, b \in \{1, \ldots, B'\}$
          $\mathcal{L} \leftarrow \frac{1}{B'} \sum_{b=1}^{B'} \left\| \mathbf{y}_{p_b}^{(n_b)} - \hat{\mathbf{y}}_{p_b}^{(n_b)} \right\|_2^2$
          $\mathbf{Z}_{n_b} \leftarrow \mathbf{Z}_{n_b} - O_{\text{split}} \left( \nabla_{\mathbf{z}_{n_b}} \mathcal{L} \right)$     ▷ In parallel $\forall\, b \in \{1, \ldots, B'\}$
        **end for**
      **end for**
    **end for**
    **return** $\mathcal{Z}_{\text{train}}, \mathcal{Z}_{\text{validation}}, \mathcal{Z}_{\text{test}}$
  **end function**

Next, we apply *Neo*MLP on the permuted inputs. Making use of the equivariance property, the output of the function applied to the permuted inputs is equivalent to the permutation of the output of the function applied to the original inputs.

$$f\left(\mathbf{P}_1 \mathbf{T}^{(0)}\right) = \mathbf{P}_1 f\left(\mathbf{T}^{(0)}\right) \tag{11}$$

Since the network is only using the output tokens in the final step as an output of the network, the overall behaviour of *Neo*MLP is invariant to the permutations of the hidden nodes.

We can follow the same principle to show that permuting the output nodes results in different outputs. The permutation matrix in this case is $\mathbf{P}_2 = \mathbf{I}_{I \times I} \oplus \mathbf{I}_{H \times H} \oplus \mathbf{P}_{O \times O}$. The equivariance property still holds, namely $f\left(\mathbf{P}_2 \mathbf{T}^{(0)}\right) = \mathbf{P}_2 f\left(\mathbf{T}^{(0)}\right)$. However, the output tokens are now used as the output of the network. This means that permuting the output tokens would result in permuting the output dimensions of a signal, which is clearly not equivalent to the original signal.

A corollary of the permutation symmetries is that if we start with a randomly initialized model, apply a permutation on the hidden nodes to create another model, and then train the two models independently, these two trained models would be identical up to the permutation of the hidden nodes. This observation is important for downstream tasks, as it shows the existence of equivalence classes that should be taken into account by the downstream models.

## C  COMPUTATIONAL COMPLEXITY

While *Neo*MLP comfortably outperforms Siren in the task of fitting high-resolution signals, it is also more computationally expensive. We quantitatively measure the computational complexity of our method using the `fvcore` library[3]. We evaluate on the "bikes" video signal, and use the hyperparameters described in Appendix E. We report the FLOPs for 1 input (*i.e.* 1 coordinate) in the forward pass. *Neo*MLP has 51.479 MFLOPs, out of which 17.83 MFLOPs correspond to the

---
[3] https://github.com/facebookresearch/fvcore

attention itself and 33.55 MFLOPs correspond to the FFNs. In the same setup, Siren (Sitzmann et al., 2020) has 3.15 MFLOPs.

Despite having a higher computational complexity compared to the baselines, *Neo*MLP can actually fit high resolution signals faster, and does so while having a smaller memory footprint, since it can make use of small batch sizes. Figure 5 shows the runtime of *Neo*MLP for fitting high-resolution signals, compared to the baselines. The $x$-axis represents wall time in seconds and the $y$-axis represents the reconstruction quality (PSNR). Table 5 shows the corresponding GPU memory and batch size, along with the total runtime for fitting high resolution signals.

Finally, despite the large difference in FLOPs, the forward pass of *Neo*MLP is almost as fast as the forward pass of Siren, considering the same batch size. Namely, we ran a full evaluation on the "bikes" signal, on an Nvidia H100 GPU, using a batch size of 32,768. *Neo*MLP takes 139.74 seconds, while Siren takes 131.01 seconds. *Neo*MLP, however, cannot fit larger batch sizes in memory, while Siren can fit as big as 1,048,576. With this batch size, Siren requires 79.18 seconds for a full evaluation.

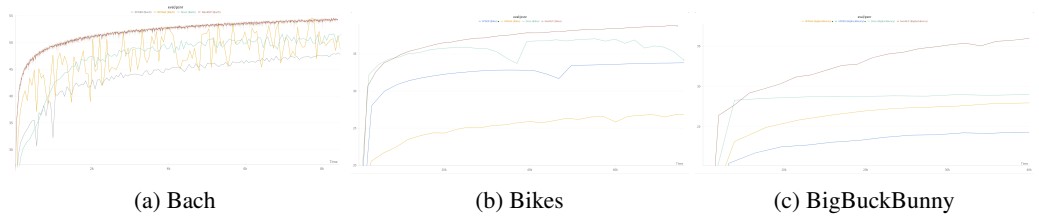

|     (a) Bach     |     (b) Bikes     |     (c) BigBuckBunny     |

Figure 5: Runtime for fitting high-resolution signals. The $x$-axis represents wall time in seconds and the $y$-axis represents the reconstruction quality (PSNR). *Neo*MLP fits signals faster and with better reconstruction quality.

Table 5: Runtime, GPU memory, and batch size on fitting high resolution signals. For each dataset, we trained all methods for the same amount of time for fair comparison.

(a) Bach

| Method | GPU memory (GB) | Batch size | Runtime (hours) |
|---|---|---|---|
| RFFNet (Tancik et al., 2020) | 3.7 | 308,207 | 2.33 |
| Siren (Sitzmann et al., 2020) | 3.9 | 308,207 | |
| SPDER (Shah & Sitawarin, 2024) | 6.0 | 308,207 | |
| *Neo*MLP (ours) | 2.2 | 4,096 | |

(b) Bikes

| Method | GPU memory (GB) | Batch size | Runtime (hours) |
|---|---|---|---|
| RFFNet (Tancik et al., 2020) | 11.2 | 262,144 | 19.07 |
| Siren (Sitzmann et al., 2020) | 16.8 | 262,144 | |
| SPDER (Shah & Sitawarin, 2024) | 37.3 | 262,144 | |
| *Neo*MLP (ours) | 11.1 | 4,096 | |

(c) BigBuckBunny

| Method | GPU memory (GB) | Batch size | Runtime (hours) |
|---|---|---|---|
| RFFNet (Tancik et al., 2020) | 13.9 | 262,144 | 24.73 |
| Siren (Sitzmann et al., 2020) | 18.7 | 262,144 | |
| SPDER (Shah & Sitawarin, 2024) | 39.2 | 262,144 | |
| *Neo*MLP (ours) | 13.2 | 4,096 | |

We also monitor the runtime of *Neo*MLP on fitting datasets of signals, and compare against Functa (Dupont et al., 2022). We report the results in Table 6. *Neo*MLP consistently exhibits lower runtimes for the fitting stage, while Functa is much faster during the finetuning stage, which can be attributed to the meta-learning employed for finetuning, and the highly efficient JAX (Bradbury et al., 2018)

implementation. As noted by Dupont et al. (2022), however, meta-learning may come at the expense of limiting reconstruction accuracy for more complex datasets, since the latent codes lie within a few gradient steps from the initialization.

Table 6: Runtime on fitting datasets of signals. The finetuning runtime is measured on the test set only. The runtime for fitting is measured in minutes, while the runtime for finetuning is measured in seconds.

(a) MNIST

| Method | Fitting | | Finetuning | |
|---|---|---|---|---|
| | Num. epochs | Runtime (min.) | Num. epochs | Runtime (sec.) |
| Functa (Dupont et al., 2022) | 192 | 240 | 3 | 16 |
| *Neo*MLP (ours) | 20 | 63 | 10 | 318 |

(b) CIFAR10

| Method | Fitting | | Finetuning | |
|---|---|---|---|---|
| | Num. epochs | Runtime (min.) | Num. epochs | Runtime (sec.) |
| Functa (Dupont et al., 2022) | 213 | 418 | 3 | 16 |
| *Neo*MLP (ours) | 50 | 305 | 10 | 646 |

(c) ShapeNet

| Method | Fitting | | Finetuning | |
|---|---|---|---|---|
| | Num. epochs | Runtime (min.) | Num. epochs | Runtime (sec.) |
| Functa (Dupont et al., 2022) | 20 | 1002 | 3 | 250 |
| *Neo*MLP (ours) | 20 | 713 | 2 | 1680 |

## D IMPLEMENTATION DETAILS

### D.1 EMBEDDING INITIALIZATION

**Fitting high-resolution signals**  We initialize input embeddings by sampling from a normal distribution with variance $\sigma_i^2 = 1$. For hidden and output embeddings, we use a variance $\sigma_o^2 = 1e-3$.

**Fitting $\nu$-sets**  During fitting, we initialize the input, hidden, and output embeddings by sampling a normal distribution with variance $\sigma_i^2 = \sigma_o^2 = 1e-3$. During finetuning, we sample embeddings for new signals from a normal distribution with variance $\sigma_o^2 = 1e-3$.

### D.2 WEIGHT INITIALIZATION

We initialize the bias of the final output linear layer to zeros, as we observe this leads to faster convergence and better stability at the beginning of training. Further, we initialize the weights of the linear projection following the random Fourier features by sampling from a normal distribution $\mathcal{N}\left(0, \frac{2}{D_{\text{RFF}}}\right)$. This results in a unit normal distribution of the inputs after the linear projection.

## E EXPERIMENT DETAILS

### E.1 HIGH-RESOLUTION SIGNALS

In Table 7 we provide the hyperparameters for *Neo*MLP.

For the audio fitting, Siren (Sitzmann et al., 2020) has 198,145 parameters. It is a 5-layer MLP, with a hidden dimension of 256, and it is trained with full batch training and a learning rate of $5 \cdot 10^{-5}$.

Table 7: Training and backbone hyperparameters for fitting high-resolution signals.

| Hyperparameter | Audio (Bach) | Video (Bikes) | Video+Audio (Big Buck Bunny) |
|---|---|---|---|
| Number of parameters | 182,017 | 3,189,249 | 3,189,249 |
| FFN hidden dim | 256 | 1,024 | 1,024 |
| Token dimensionality $D$ | 64 | 256 | 256 |
| Self-attention heads | 4 | 8 | 8 |
| Number of layers | 3 | 4 | 4 |
| RFF dimensionality $D_{\text{RFF}}$ | 512 | 128 | 128 |
| RFF standard deviation $\sigma$ | 20 | 20 | 20 |
| Total number of nodes | 8 | 16 | 16 |
| Number of epochs | 5,000 | 200 | 400 |
| Minibatch size | 4,096 | 4,096 | 4,096 |
| Learning rate | 0.005 | 0.0005 | 0.0005 |

For the video fitting, Siren has 3,155,971 parameters, and for the audio-visual data, Siren has 3,162,121 parameters. It both settings, it is using the exact same architecture with 5 layers and a hidden dimension of 1024. We train it with a learning rate of $10^{-4}$ and a batch size of 262,144.

### E.2 FITTING $\nu$-SETS

For ShapeNet10 (Chang et al., 2015), we fit the dataset for 20 epochs. In each epoch, we stop when we have used 10% of the available points, which effectively results in 2 epochs in total. We finetune for 2 epochs, and use the 20% of the available points. We use a minibatch size of 32,768 points, and a learning rate of 0.005. The hyperparameters of the backbone are listed in Table 8.

For MNIST, we fit the dataset for 20 epochs and finetune for 10 epochs. We use a minibatch of 12,288 points (the equivalent of 16 images), and a learning rate of 0.005. The hyperparameters of the backbone are listed in Table 8.

For CIFAR10, we fit the dataset for 50 epochs and finetune for 20 epochs. We use a minibatch of 16,384 points (the equivalent of 16 images), and a learning rate of 0.005. The hyperparameters of the backbone are listed in Table 8.

Table 8: Training and backbone hyperparameters for ShapeNet10, MNIST, and CIFAR10

| Hyperparameter | ShapeNet10 | MNIST | CIFAR10 |
|---|---|---|---|
| Training Hyperparameters | | | |
| Minibatch size | 32,768 | 12,288 (16 images) | 16,384 (16 images) |
| Learning rate | 0.005 | 0.005 | 0.005 |
| Backbone Hyperparameters | | | |
| FFN hidden dim | 512 | 512 | 128 |
| Token dimensionality $D$ | 256 | 256 | 512 |
| Number of self-attention heads | 4 | 4 | 4 |
| Number of layers | 3 | 3 | 3 |
| RFF dimensionality $D_{\text{RFF}}$ | 512 | 512 | 128 |
| RFF standard deviation $\sigma$ | 20 | 20 | 20 |
| Total number of nodes | 8 | 8 | 8 |

### E.3 DOWNSTREAM TASKS ON $\nu$-SETS

We perform a hyperparameter search for *Neo*MLP to find the best downstream model. Specifically, we use Bayesian hyperparameter search from Wandb (Biewald, 2020) to find the best performing hyperparameters for CIFAR10, and reuse these hyperparameters for all datasets. We perform our

search over the choice of Mixup (Zhang, 2017), batch size, learning rate, noise added to the data, data dropout, hidden dimension and model dropout (Srivastava et al., 2014).

Our downstream model is a 3 layer MLP with SiLU activations (Ramachandran et al., 2018), a hidden dimension of 2048, and dropout of 0.3. We train the model with a learning rate of $8e-3$, and batch size of 256. We use Mixup, weight decay with $\lambda = 0.05$, and add noise to the data with scale 0.05. Finally, we use weight averaging with exponential moving average (EMA).

For CIFAR10 (Krizhevsky et al., 2009), the model takes as input 6 embeddings (the *Neo*MLP had 8 nodes in total). We train for 100 epochs.

For ShapeNet10 (Chang et al., 2015), the model takes as input 13 embeddings (the *Neo*MLP had 16 nodes in total). We use a higher weight decay $\lambda = 0.25$ to further prevent overfitting, and train for 500 epochs.

For MNIST (LeCun et al., 1998), the model takes as input 6 embeddings (the *Neo*MLP had 8 nodes in total). We use a higher weight decay $\lambda = 0.2$ and train for 500 epochs.

## F  DATASET DETAILS

### F.1  SHAPENET10

We use the following 10 classes for ShapeNet10 classification: loudspeaker, bench, watercraft, lamp, rifle, sofa, cap, airplane, chair, table.

The dataset comprises 35,984 shapes. We use 29,000 shapes for training, 2,000 as a validation set, and 4,984 as a test set.

For CIFAR10, following Functa (Dupont et al., 2022), we use 50 augmentations per training and validation image. This results in a total of 2,500,000 training and validation images. We use 5,000 of those for validation.

## G  QUALITATIVE RESULTS

We show example frames for the "BigBuckBunny" video clip in Figure 6.

We show the reconstructions for the "Bach" audio clip in Figure 7, and the errors between the groundtruth signal and reconstructions in Figure 8.

## H  VISUALIZATIONS

## I  LATENT SPACE

We visualize the learned MNIST data manifold for a two-dimensional latent space (with a single embedding) in Figure 10, following Kingma & Ba (2015); Park et al. (2019). We assume that the latent space is Gaussian, with a sample mean and variance estimated from the latents of the training set. We sample linearly spaced coordinates on the unit square and transform them through the Percent Point Function (PPF) of the Gaussian to produce the values of the latent variables.

We also visualize random samples from the latent space of *Neo*MLP across hyperparameter configurations varying in the number of embeddings and dimensionality of the latents in Figure 11, as well as the corresponding reconstruction quality and downstream performance in Table 9.

## J  MORE ABLATION STUDIES

We report further ablation studies on CIFAR10 (Krizhevsky et al., 2009) to examine the importance of various hyperparameters in *Neo*MLP; namely, we perform ablation studies on the number of layers, the number of hidden latents, and using different layer normalization strategies. In all ablation studies, we fit the dataset for 20 epochs and finetune for 5 epochs. The remaining hyperparameters (except for the target variable in each study) are identical to the CIFAR10 column in Table 8 in Appendix E.2.

Table 9: Reconstruction quality and downstream performance for the configurations corresponding to Figure 11.

| Num. latents | Latent dim. | Fit PSNR ($\uparrow$) | Accuracy (%) |
|---|---|---|---|
| 1 | 2 | 15.61 | 23.7 |
| 1 | 8 | 20.47 | 48.2 |
| 2 | 4 | 20.19 | 49.3 |
| 1 | 32 | 24.75 | 76.2 |
| 4 | 8 | 24.44 | 72.5 |

Table 10: Ablation study on the importance of the number of layers on CIFAR10. Using 4 layers results in the best test reconstruction quality as well as the best downstream performance. Increasing the number of layers to 8 marginally increases the fitting PSNR, while negligibly reduces the test PSNR and the test accuracy.

| Num. layers | Fit PSNR ($\uparrow$) | Test PSNR ($\uparrow$) | Accuracy (%) |
|---|---|---|---|
| 1 | 23.10 | 24.04 | 53.92 |
| 2 | 29.54 | 28.76 | 57.36 |
| 3 (default) | 30.54 | 29.15 | 58.82 |
| 4 | 33.51 | **31.15** | **59.09** |
| 8 | **34.67** | 31.05 | 58.96 |

Table 11: Ablation study on the importance of layer normalization on CIFAR10. Our choice of removing the Layer Norm (Ba et al., 2016) is backed by this ablation study, since it results in the best test PSNR and accuracy. Both Layer Norm and RMS Norm (Zhang & Sennrich, 2019) achieve good fitting PSNR, but fail to generalize in the test set.

| Normalization type | Fit PSNR ($\uparrow$) | Test PSNR ($\uparrow$) | Accuracy (%) |
|---|---|---|---|
| No normalization (default) | 30.54 | **29.15** | **58.82** |
| Layer Norm (Ba et al., 2016) | **30.82** | 26.76 | 46.51 |
| RMS Norm (Zhang & Sennrich, 2019) | 29.96 | 26.16 | 44.11 |

Table 12: Ablation study on the importance of the number of hidden latents ($H$) on CIFAR10. The total number of latents is the sum of the hidden and output latents. Increasing the number of latents results in increasing reconstruction quality, at the expense of lower downstream performance. Perhaps surprisingly, the best downstream performance is acquired without hidden latents; we hypothesize that this can be attributed this to the lack of permutation symmetries that stems from using output latents only.

| $H$ | Fit PSNR ($\uparrow$) | Test PSNR ($\uparrow$) | Accuracy (%) |
|---|---|---|---|
| 0 | 30.49 | 29.40 | **60.64** |
| 3 | 30.54 | 29.15 | 58.82 |
| 11 | **40.05** | **35.40** | 57.60 |

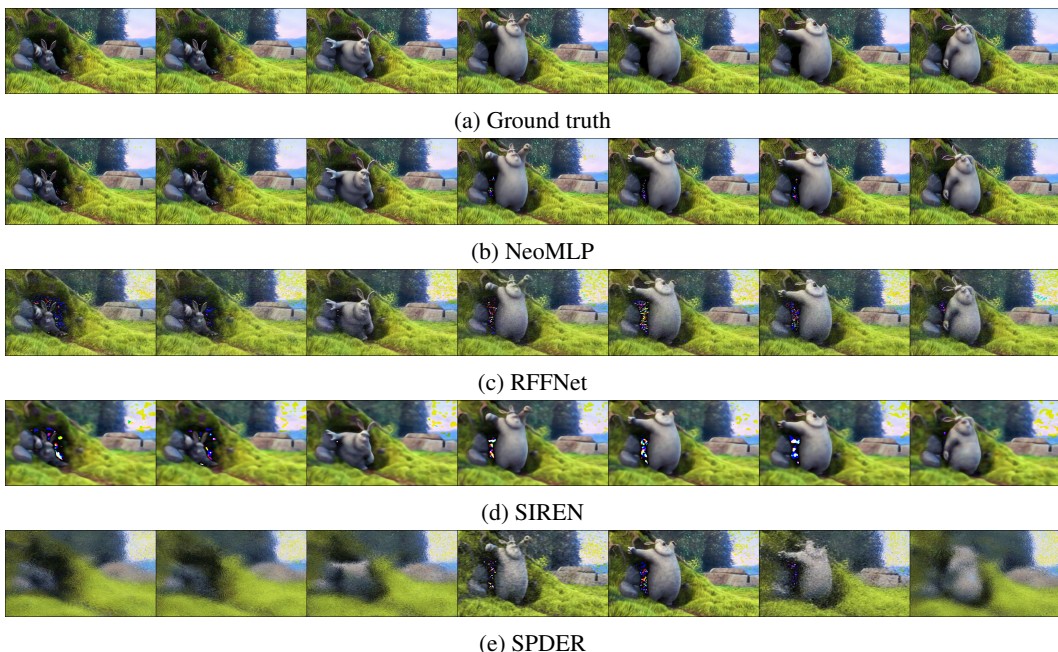

(a) Ground truth

(b) NeoMLP

(c) RFFNet

(d) SIREN

(e) SPDER

Figure 6: Examples frames from fitting the "BigBuckBunny" video clip. The first row shows the groundtruth, while the following rows show the reconstructions obtained using *Neo*MLP, RFFNet, Siren, and SPDER, respectively. We observe that *Neo*MLP learns to reconstruct the video with much greater fidelity.

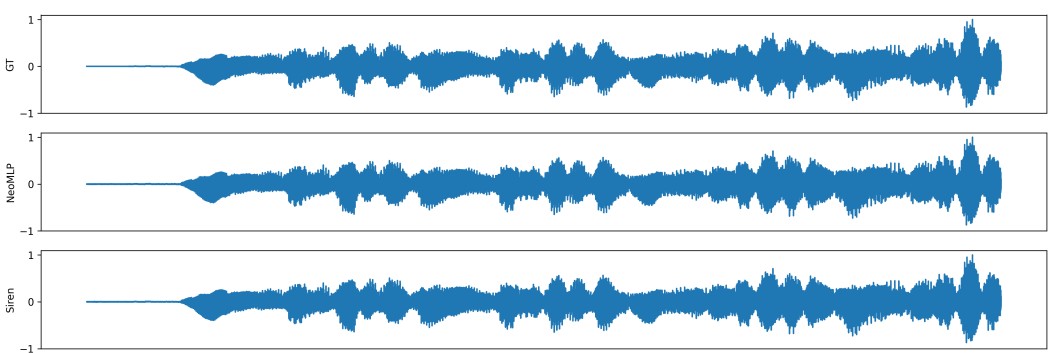

Figure 7: Predictions for the "Bach" audio clip. The first row shows the groundtruth signal, while the second and third row show the reconstructions from *Neo*MLP and Siren, respectively.

**Importance of RFF**    As shown by Rahaman et al. (2019), neural networks suffer from *spectral bias*, *i.e.* they prioritize learning low frequency components, and have difficulties learning high frequency functions. We expect that these spectral biases would also be present in NeoMLP if left unattended. To that end, we employed Random Fourier Features (RFF) (Tancik et al., 2020) to project our scalar inputs to higher dimensions. Compared to alternatives like sinusoidal activations (Sitzmann et al., 2020), RFFs allow our architecture to use a standard transformer.

To examine the spectral bias hypothesis, we train *Neo*MLP without RFF, using a learnable linear layer instead. We train this new model on the "bikes" video, and on MNIST. We present the results in Table 13. The study shows that RFFs clearly help with reconstruction quality, both in reconstructing a high-resolution video signal, and on a dataset of images. Interestingly, the reconstruction quality drop from removing RFFs does not translate to downstream performance drop, where, in fact, the model without Fourier features is marginally better than the original.

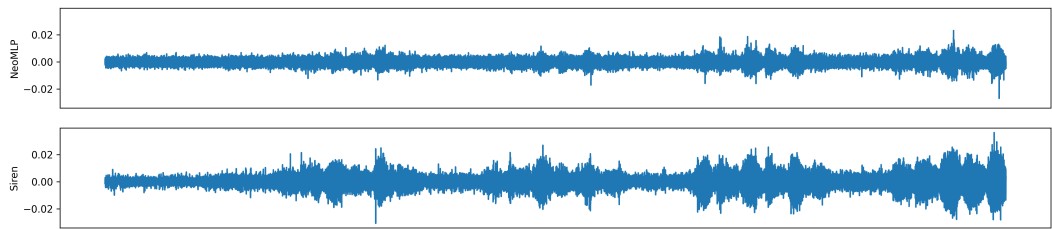

Figure 8: Errors $\boldsymbol{\epsilon} = \mathbf{y} - \hat{\mathbf{y}}$ between predictions $\mathbf{y}$ and groundtruth $\hat{\mathbf{y}}$. The top row shows the error for *Neo*MLP, while the bottom row shows the error for Siren. Both the $x$-axis and the $y$-axis are shared in this figure, *but* the $y$-axis is different from Figure 7 . We see that the errors from Siren have a much larger amplitude, and still seem to capture signal components.

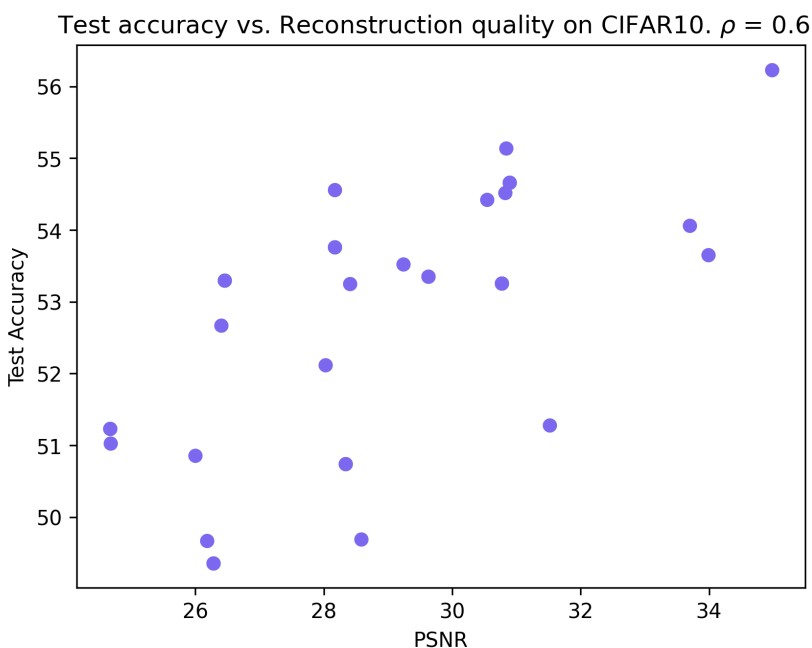

Figure 9: Test accuracy vs. reconstruction quality (PSNR). Experiments on CIFAR10, with different hyperparameters, *without* augmentations.

Table 13: Ablation study on the importance of random Fourier features on (a) the bikes video, (b) on MNIST.

| (a) "Bikes" video | | (b) MNIST | | |
|---|---|---|---|---|
| Method | PSNR (↑) | Method | PSNR (↑) | Accuracy (%) |
| *Neo*MLP (without RFF) | 35.92 | *Neo*MLP (without RFF) | 30.33 | $\mathbf{98.81}_{\pm 0.03}$ |
| *Neo*MLP | **39.06** | *Neo*MLP | **33.98** | $98.78_{\pm 0.04}$ |

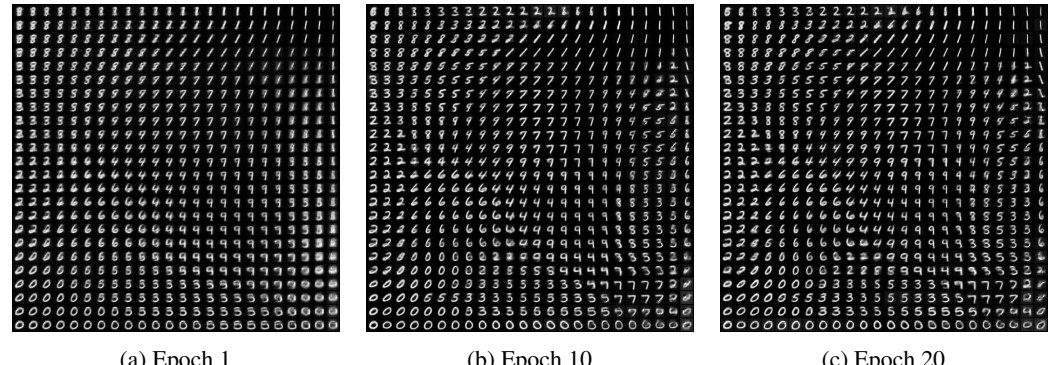

(a) Epoch 1      (b) Epoch 10      (c) Epoch 20

Figure 10: The data manifold of *Neo*MLP with a 2D latent space and a single embedding, *i.e.* $O = 1, H = 0, D = 2$. We visualize the manifold as the fitting stage progresses.

(a) $H = 0, O = 1, D = 2$

(b) $H = 0, O = 1, D = 8$

(c) $H = 1, O = 1, D = 4$

(d) $H = 0, O = 1, D = 32$

(e) $H = 3, O = 1, D = 8$

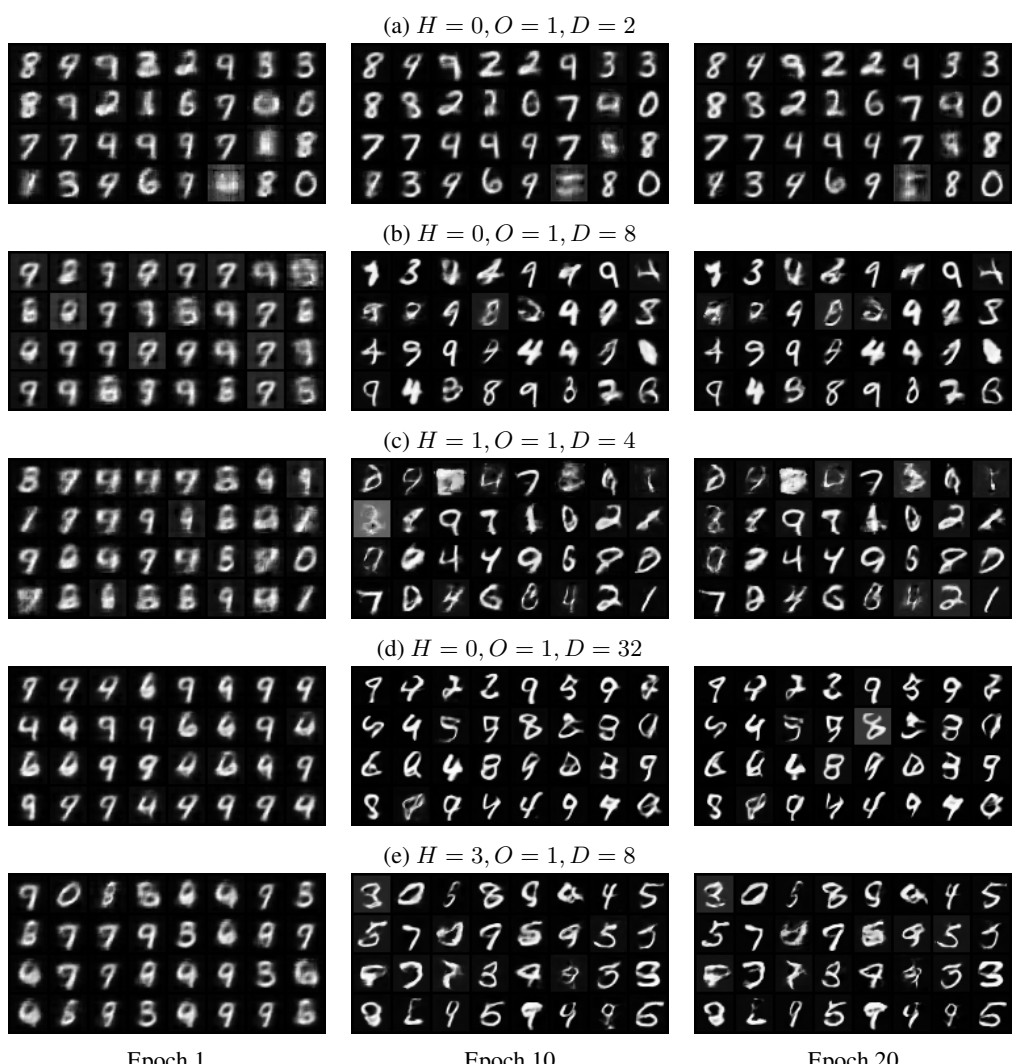

Epoch 1      Epoch 10      Epoch 20

Figure 11: Random samples from the latent space of *Neo*MLP, as the fitting stage progresses. We visualize various configurations of the number of embeddings and the dimension of the latents.

