# OpenReview forum: "From MLP to NeoMLP: Leveraging Self-Attention for Neural Fields"
_ICLR.cc/2026/Conference — ICLR 2026 Conference Withdrawn Submission_

### Official Review · Reviewer_nRsc · 2025-10-26

**Soundness:** 1
**Presentation:** 2
**Contribution:** 1
**Rating:** 2
**Confidence:** 4

**Summary:**

This paper proposes NeoMLP, an architecture motivated by the view that an MLP is a graph and could be processed like a graph, such as via message passing. NeoMLP does message passing by using self-attention between input, hidden, and output nodes. The authors then use NeoMLP as a conditional neural field (NeF). NeoMLP is evaluated on fitting signals and creating neural representations for datasets (MNIST, CIFAR-10, ShapeNet-10) and evaluating these neural representations on a downstream task (classification accuracy).

**Strengths:**

This paper proposes a new type of neural field based on self-attention. The method’s choices are justified through ablations. The paper’s reproducibility is also good, as many of the hyperparameters and training details are provided as well as the code.

**Weaknesses:**

**(W1) Contributions relative to other work**: This paper does not acknowledge or discuss the similarities between the proposed NeoMLP architecture and past work that is similar, such as transformers and transformers with registers.

**(W2) Novelty**: As I understand it, NeoMLP is essentially a transformer with registers with minor tweaks, being used as an autodecoding NeF. As such, the novelty of this work is low.

**(W3) Choice of baselines**: This paper is missing the latest literature on NeF architectures, comparing only to RFFNet, SIREN, and SPDER. The first two are about 5 years old at this point. While SPDER is recent (2024), there is no claim that SPDER is better than all the other NeF architectures that have been published in the intervening time period.

Furthermore, this paper ignores much of the literature of what this paper’s terminology describes as “neural dataset fitting”, which seeks to produce neural representations for whole datasets of signals. In particular, this paper does not compare to alternative methods using meta-learning or hypernetworks. These works are also missing from the related works.

**(W4) Empirical results**: First, this paper only evaluates on simple tasks, namely signal fitting and downstream classification accuracy. More complicated tasks typically done using NeFs, such as novel view synthesis, are not investigated. The datasets are also not very complex (for Table 1, only 3 total example signals are fit) and are generally low resolution.

Also, it is quite strange that in Table 2, DWSNet, Neural Graphs, and Fit-a-NeF all have the same PSNR and IoU, but significantly different downstream classification accuracies. For example, while DWSNet and Fit-a-NeF have the same PSNR, their accuracies differ substantially (85.70% vs 96.40%). On the ShapeNet dataset, DWSNet, Neural Graphs, and Fit-a-NeF all have the same PSNR despite having different accuracies that also differ substantially (91.06% vs 90.31% vs 82.96%). These accuracies are all lower than Functa, but all these methods have the same higher PSNR. Furthermore, the PSNRs and IoUs reported as coming from the Fit-a-NeF cannot be found in either the main text or supplementary material of that paper, although the accuracies of these are reported in the Fit-a-NeF paper.

Furthermore, there is no comparison between qualitative examples on the datasets in Table 2 (MNIST, CIFAR-10, ShapeNet-10), which would help contextualize the reported numerical results.

**(W5) Computational complexity**: Since NeoMLP is essentially a transformer, it requires much more computational methods than MLP-based NeFs. Additionally, fine-tuning is much more expensive because of the need to fit on all splits of the dataset. NeoMLP also necessitates storing the training and validation sets along with the model, due to the fact that they need to be re-fit during inference.

**(W6) Method**: Why certain parts of the method are the way they are is confusing and would benefit from further explanation and justification. During NeoMLP training (Algorithm 1), the learned input, hidden, and output embeddings are discarded for the training set. Subsequently, in NeoMLP fine-tuning, these learnable embeddings need to be re-learned for the training, validation, and test sets. Normally in the autodecoding setting, the learned latents for the training set would be kept for inference and not re-trained. Secondly, it’s not clear why fitting needs to happen using all splits and not just the test dataset. It’s also not clear why the fitting happens sequentially each epoch with different optimizers for each split. It also doesn’t seem necessarily fair nor reasonable to assume that the training and validation sets are available at inference time.

**Questions:**

**(Q1)**: How does NeoMLP compare to recent MLP-based NeF architectures other than SPDER?

**(Q2)**: Why do DWSNet, Neural Graphs, and Fit-a-NeF have the same PSNR and IoU on the MNIST, CIFAR-10, and ShapeNet-10 datasets (Table 2) despite being different methods and having significant differences in accuracy?

**(Q3)**: If there is time, how does NeoMLP perform on more complex datasets such as CIFAR-100 or ImageNet?

**(Q4)**: Why is SIREN faster when using the maximum batch size than when using a medium-sized batch size (c.f. Appendix C)?

**(Q5)**: In Algorithm 2, why do latent embeddings for the training and validation sets need to be re-trained? Why is there a requirement that all splits of the dataset be used? Why does each split need its own optimizer? What happens if fitting is only done on the test set (as is normally the case)?

**(Q6)**: Normally, during inference time, only the test set is used. Why is it reasonable to assume that the training and validation sets are also available?

---

### Official Review · Reviewer_LiLc · 2025-10-31

**Soundness:** 2
**Presentation:** 3
**Contribution:** 2
**Rating:** 4
**Confidence:** 4

**Summary:**

The paper introduces:
1. NeoMLP which is a graph MLP that shared weight between nodes using self-attention mechansim

**Strengths:**

1. The method is able to outperform the baseline with the same number of parameters.

**Weaknesses:**

1. The paper failed to elaborate further on how exactly are the weights shared between in the MLP.
2. In the abstract, it was written that NeFs fails at downstream task like classification and segmentation but there is no results on segmentation task.

**Questions:**

1. The paper Describes NeoMLP as a graph using self-attention. But if one were to expand the terms we would get $h_i^{(l-1)} \sum_j \(W_Q^{(l)})^T W_K^{(l)}W_V^{(l)} (h_j^{(l-1)})^2$ this would be just linear layers?
2. Could you elaborate more on the connectivity of the graph? I am confused as  to where the complete graph is being handled?
3. How are the gradients handled during back-propagation if it is a full connected graph?

---

### Official Review · Reviewer_2NU4 · 2025-10-31

**Soundness:** 3
**Presentation:** 3
**Contribution:** 2
**Rating:** 4
**Confidence:** 4

**Summary:**

The paper presents NeoMLP, a novel implicit neural representation (INR) framework designed to model diverse signal types (e.g., images, audio) through a unified architecture. The core idea is to reinterpret an MLP as a fully connected graph where all nodes (input, hidden, output) communicate via self-attention. Each signal instance has its own latent codes (hidden and output embeddings), which are optimised jointly with shared network parameters.

The method is evaluated on two tasks: (1) signal representation, via PSNR on audio, video, and multimodal data; and (2) downstream classification, using the learned latent codes as features.

The paper’s main strength lies in its conceptual originality. It combines INRs, attention-based message passing, and structured latent conditioning into a coherent framework. However, the practical value and quantitative advantages over established approaches remain uncertain. The current results demonstrate fitting capacity but not clear superiority or applicability in demanding real-world tasks. The study convincingly motivates a new conceptual direction, but the evidence of its practical competitiveness is still limited.

**Strengths:**

**S1: Architectural novelty.** NeoMLP redefines the MLP through a fully connected, attention-based graph interpretation. This is an imaginative and promising step toward bridging MLP-style neural fields and transformer-based representations.

**S2: Unified multi-modal representation.** Demonstrating that a single INR can fit both audio and video signals is conceptually strong and suggests extensibility to other modalities.

**S3: Clarity and presentation.** The manuscript is well written, logically structured, and carefully contextualised within prior INR and graph-based literature. The theoretical motivation is clear, and connections to related work are properly acknowledged.

**Weaknesses:**

**W1: Limited practical validation.** The paper convincingly demonstrates that NeoMLP can fit complex signals but does not establish a clear advantage in realistic applications. The classification experiments on MNIST, CIFAR-10, and ShapeNet10 offer only limited insight into how the learned representations compare with established self-supervised features. For instance, it remains unclear how NeoMLP latents would perform against CLIP or DINO embeddings in an equivalent MLP-based classification setup. Without such benchmarks, the work provides no evidence of broader gains in efficiency, accuracy, or scalability beyond the INR domain. Nor does it show that the model improves the efficiency of practical INR fitting, such as when serving as a backbone for neural rendering tasks like NeRF.

**W2: No analysis of latent space geometry.** The paper does not examine how the learned representations behave in terms of interpolation, smoothness, or semantic structure. Prior INR works (e.g., DeepSDF [1], IGR [2], and, generally, VAE-style learning models) provide insights into geometric and semantic continuity; such analysis would significantly strengthen the paper’s claim of representational power.

**W3: Unclear evaluation procedure.** The PSNR evaluation appears to measure reconstruction over all coordinates of a signal, including those seen during training. Because coordinates are sampled with replacement, there is no explicit test set of unseen pixels or timesteps. Consequently, PSNR reflects fitting fidelity rather than genuine generalisation or extrapolation, and this should be clarified.


References

[1] Park et al., DeepSDF: Learning Continuous Signed Distance Functions for Shape Representation, CVPR 2019.

[2] Gropp et al., Implicit Geometric Regularization for Learning Shapes, ICML 2020.

**Questions:**

**Q1. Practical advantage**: What is the main expected benefit of NeoMLP representations compared with e.g. state-of-the-art alternatives such as CLIP or DINO? For instance, how do downstream results on CIFAR-10 or MNIST compare when using these features?

**Q2. Extension to NeRF**: Could NeoMLP’s structure help in learning NeRF-style scene representations, and if so, what efficiency or modelling benefits would it provide?

**Q3. Latent interpolation**: Have the authors explored interpolation between latent codes of different signals? Does this yield smooth or semantically meaningful transitions in the reconstructed outputs?

---

### Official Review · Reviewer_nwJF · 2025-11-03

**Soundness:** 3
**Presentation:** 3
**Contribution:** 2
**Rating:** 2
**Confidence:** 4

**Summary:**

This paper proposes NeoMLP, a transformer-like architecture that re-interprets a Multi-Layer Perceptron (MLP) as a fully connected message-passing graph. Each node (input, hidden, output) carries high-dimensional features, and self-attention replaces edge-specific weights for scalability. Hidden and output nodes serve as latent codes, enabling NeoMLP to act as a conditional neural field (NeF) capable of representing complex and multimodal signals. The authors introduce **$\nu$-reps** for latent-code representations learned per instance. And they demonstrate their utility on downstream classification tasks (MNIST, CIFAR-10, ShapeNet10) with the **$\nu$-sets** (representations of instances in the dataset)

**Strengths:**

1. Viewing an MLP as a graph and replacing layerwise propagation with self-attention-based message passing is conceptually good. This formulation bridges MLPs, transformers, and graph neural networks.

2. NeoMLP surpasses Siren, RFFNet, and SPDER for high-resolution and multimodal signal fitting (audio, video, audio-visual) with up to +10 dB PSNR improvement, especially for representing videos. The **$\nu$-sets** setting for downstreaming task such as classification also shows the advantage of this neural representations.

**Weaknesses:**

1. The idea of treating MLPs as graphs and performing message passing has been explored (e.g., Nikolentzos et al. 2024; Lim et al. 2024; etc). The main novelty that replaces edges with self-attention is incremental, and the name “NeoMLP” may oversell the change.

2. When fitting signals to learn neural representations, the capacity of the network significantly influences the reconstruction quality. This paper demonstrates large improvements on reconstruction PSNR, but it is know that self-attention layers are much heavier than the computation cost of MLPs. So it is important to compare in similar computation budges (e.g., GMACs, or similar number of model parameters).

3. The paper could more directly compare NeoMLP to transformer-based NeRFs (e.g., Scene Representation Transformer, Perceiver IO). Without such baselines, it is difficult to gauge whether the benefits stem from the specific graph-MLP reinterpretation or from using standard attention machinery.

**Questions:**

1. How sensitive are results to the number of hidden tokens and attention heads?

2. Please report and compare the computational complexity, including GMACs and the number of model parameters used.

---

### Note · Authors · 2025-12-03

**Comment:**

We would like to thank the reviewers for their time and constructive criticism. We will make use of the valuable comments to revise our work for future submissions.

**Withdrawal Confirmation:**

I have read and agree with the venue's withdrawal policy on behalf of myself and my co-authors.